# Development of a New Additive Based on Textile Fibers of End-of-Life Tires (ELT) for Sustainable Asphalt Mixtures with Improved Mechanical Properties

**DOI:** 10.3390/polym14163250

**Published:** 2022-08-10

**Authors:** Gonzalo Valdés-Vidal, Alejandra Calabi-Floody, Carla Duarte-Nass, Cristian Mignolet, Cristian Díaz

**Affiliations:** 1Department of Civil Engineering, Universidad de La Frontera, Temuco 4811230, Chile; 2CDI—Centro de Desarrollo e Investigación, Maipú 9260061, Chile

**Keywords:** asphalt mixture, additive, end-of-life tires (ELT), performance properties

## Abstract

End-of-life tires (ELT) are a worldwide problem. Rubber, steel, and different textile fibers are the by-products of ELT. Unlike rubber and steel, waste tire textile fibers (WTTF) are disposed of in landfills or burned. This study developed an additive made with WTTF to be incorporated into conventional hot mix asphalt (HMA), and its performance properties were evaluated. First, a characterization of the WTTF used was made and a manufacture protocol was established. Then, a reference HMA was designed and mixtures with different addition percentages (2%, 5% and 8%) of the WTTF-based additive were evaluated. The mechanical properties studied were stiffness modulus, moisture susceptibility, rutting resistance, stripping, and cracking resistance. The results indicated that the addition of the 2% and 5% WTTF-based additive improved these performance properties. Moreover, all addition percentages of the WTTF-based additive evaluated demonstrated a decrease of over 29% in permanent deformation according to the Hamburg Wheel Tracking Test. Thus, the use of the WTTF would not only be valuing a waste, but an asphalt mixture with improved properties would be obtained, contributing to the circular economy by reusing a material and prolonging the useful life of the asphalt mixture.

## 1. Introduction

Every year around the world about 1.5 billion end-of-life tires (ELT) are generated [1], representing almost 20.6 million metric tons in 2018 [2]. To face this situation, the major tire producers mainly reuse or recycle them, with the European Union being the leader of this practice with 92% recycled or reused tires [3]. However, in developing regions such as Latin America, most ELT are disposed of in landfills or in unlawful sites, causing environmental and health problems, and the recovered ELT are mainly used as a supplementary fuel in cement kilns [1].

Tire compositions differ depending on the application, either for light vehicles or trucks and heavy machinery. Due to their complex composition, tires are difficult to recycle and the use of their materials requires different separation processes based on size reduction [2,4]. The main materials manufactured from the recycling of tires are rubber (granulated or powder), steel cords, and polymer fibers, representing 70%, 5–30%, and 5–15% by weight, respectively [5,6,7]. The rubber obtained from recycled tires is applied in hot mix asphalts, plastic compounds, concrete, rails, and athletics tracks [6,8]. Special attention is paid to pyrolysis processes to recover both energy and materials (liquid or gas hydrocarbon compounds and carbon-rich solids) [1]. Tire rubber is applied in hot mix asphalt because it contributes to higher performances for noise reduction, aging resistance, high-temperature stability, and fatigue resistance [9]. The recovered steel could be applied as an additive for cement reinforcement or be reused as reprocessed steel [6]. However, fibers recovered from ELT are difficult to use directly in any application because they are contaminated with rubber particles (15–60% by weight) [5,7,10,11] and a cleaning process is required. In addition, their low density makes their transportation expensive [8]. Therefore, these textile fibers are commonly sent to landfills or incinerators [6,12].

Just a few years ago, some studies focused on waste tire textile fiber (WTTF) recycling in different fields, such as pavement engineering, soil reinforcement, construction of moisture barriers in landfills, sound absorber products, reinforcement in plastic compounds, low-cost organic solvents sorbent compounds in soils treated with cement, artificial grass, shoe soles, and clothing [6,13].

The textile fibers commonly used in tire manufacturing are polyester, polyamide (nylon 6 or nylon 6.6), and rayon [6,12]. These polymer fibers, along with steel, are the basis for providing tensile strength in tires [5], which is important to the fiber content in tires for light vehicles and the steel content in heavy machinery tires. This characteristic makes WTTF suitable for reinforcing different matrixes such as cement or asphalt mixes.

In hot mix asphalt (HMA), different types of fibers are used for various purposes. Mainly, cellulose and mineral fibers are used in porous asphalt mixtures and stone mastic asphalt (SMA) mixtures to avoid bitumen drain down [6]. Table 1 shows some studies that test different fibers applied as additives in asphalt mixes. Considering nylon and polyester fibers shown in Table 1, these materials showed good results in Marshall stability and in resistance to permanent deformation and fatigue behavior, which indicates that WTTF additives made of those materials could have the same results. Moreover, Bocci and Prosperi (2020) and Landi et al. (2018) demonstrated that WTTF showed enhanced resistance to fatigue and, additionally, increased indirect tensile strength [6,8].

In this context, this study sought to develop an additive made using WTTF applied to conventional HMA to improve its performance properties. First, the WTTF raw material was characterized, and a standard manufacturing procedure was established. Then, a reference asphalt mixture was designed, which was used to evaluate three different addition percentages of the WTTF-based additive in the HMA. Finally, the properties of the asphalt mixture related to its performance in pavement, namely stiffness modulus, moisture susceptibility, rutting resistance, stripping, and cracking resistance, were evaluated for each addition percentage of the WTTF-based additive selected.

## 2. Materials and Methods

### 2.1. Development of the WTTF-Based Granular Additive

#### 2.1.1. Characterization of the Materials That Compound the Additive

##### Textile Fiber

The WTTF used in this study was obtained from a local company. The WTTF was analyzed by variable pressure scanning electron microscopy (SEM) with a STEM SU-3500 transmission module (Hitachi, Tokyo, Japan) and QUANTAX 100 energy dispersive X-ray (EDX) spectrometer detector (Bruker, Berlin, Germany) for semi-quantitative analysis. The accelerating voltage used for image processing was 15 kV and the scale grade used was 200 µm. The dispersion of each chemical element is shown in Figure 1. The semi-quantitative analysis indicated that the composition is mainly carbon (83.2%) and oxygen (12.6%), and small fractions of sulfur (1.9%), silicon (1.1%), iron (1.0%) and sodium (0.2%). The absence of nitrogen in the sample is evidence that the WTTF provided by the local company was not a nylon compound, since that polymer is based on polyamide chains. The elemental composition suggested that the main polymer of this fiber is polyester, as described by Bocci and Prosperi (2020) [6].

In addition, a thermogravimetric analysis (TGA) was carried-out in a Simultaneous Thermal Analyzer (STA) 6000 (PerkinElmer, Waltham, MA, USA), to identify the melting point of the WTTF. The TGA temperature profile used in the analysis was as follows: nitrogen gas flow at 50 mL/min, heat from 25 °C to 600 °C at 15 °C/min. Figure 2a shows the temperature profile from the TGA. The melting point of the WTTF was measured at 241.75 °C. This melting point is consistent with the melting range identified for recycled polyester [28,29]. As HMA usually applies a process temperature between 140° to 190 °C, WTTF would present a stable behavior since its melting point is higher than that temperature gap.

To confirm the polyester composition of the WTTF used in this study, an analysis of the functional groups present in the fiber was made by Fourier transform infrared spectrometry (FTIR) in a PerkinElmer FTIR Spectrum Two model with attenuated total reflectance (ATR) system. Figure 2b shows the FTIR results, which indicated signs in the 1711, 1244.14, and 871.4 cm^−1^ bands, corresponding to the terephthalate group and aromatic rings, respectively. These are characteristic signs of PET-type polyester, which agrees with the results reported by Bocci and Prosperi (2020) [6]. From the previously described results, the WTTF used in this study correspond to polyester fiber. Polyester is a durable thermoset polymer, which is being studied to be applied in advanced engineering including shielding composites for electromagnetic interference, energy conversion devices, textile materials, and biomedical devices [30].

##### Asphalt Binder

To obtain a granular additive, which facilitates the incorporation of the fiber into the asphalt mixes, a cationic asphalt emulsion was chosen as the binder. The cationic asphalt emulsion used was CRS-2, which conforms to Chilean specifications [31]. It is a rapid setting cationic emulsion, with 65% asphalt binder (AB), and it has a degree of penetration greater than 90 dmm at 25 °C. In addition, this AB is a versatile emulsion with good compatibility and adhesion onto aggregates, used mainly for the application of a single or double chip seal treatment.

##### Non-Stick Agent

A non-stick agent was required to be incorporated into the granular additive to avoid the agglomeration of the additive pellets. The non-stick agent used was rubber powder, another by-product of tire recycling, with a particle size smaller than 0.18 mm. A TGA for the rubber powder indicated that the sample had a decomposition temperature range of 331.28–485.51 °C. This temperature range agrees with Razali et al. (2020) and Martínez-Barrera et al. (2020), who reported decomposition temperatures of 373 °C and 395 °C, respectively [32,33]. As the decomposition temperature of rubber powder was higher than the manufacturing temperature required for HMA, this material was adequate for the manufacturing of the granular additive.

#### 2.1.2. Manufacturing Methodology for the WTTF-Based Granular Additive

After dosage assays, where the emulsion breaking time was evaluated, the manufacture of the granular additive was carried out as follows: first, disaggregated WTTF was mixed with the diluted asphalt emulsion at a weight ratio of 1:2, until reaching a homogeneous distribution of the asphalt emulsion in the fibers. Second, the WTTF–asphalt emulsion mix was dried to eliminate the water. Third, the WTTF–asphalt binder was pelletized. Finally, the pellets were mixed with rubber powder at a weight ratio of 20:1. The final weight composition of the manufactured WTTF-based granular additive was WTTF 58%, asphalt binder 37%, and rubber powder 5%. Figure 3 shows images of the WTTF-based additive and its components.

### 2.2. Experimental Study

The purpose of this section is to show the effect of the use of the developed additive on the design and mechanical or performance properties of the asphalt mixture. The experimental plan carried out to evaluate the effect of the additive developed in the asphalt mixture can be observed in Figure 4. Three experimental stages are shown. (1) The first stage considers the characterization of materials and the design of a reference asphalt mixture (HMA/R), which is usually used in surface course. (2) The second stage considers the selection of three additive addition percentages to be evaluated according to the literature review, together with determining the effect of the selected percentages on the design properties of the asphalt mixture. (3) The third stage considers the evaluation of the properties of the asphalt mixture related to its performance in pavement. Additionally, a statistical analysis was carried out to determine the significance between the results obtained according to the variables evaluated.

#### 2.2.1. Materials and Reference Asphalt Mix Design

The asphalt binder used in this study corresponds to a CA-24 conventional binder classified according to Chilean standard. The aggregates were of fluvial origin and were mainly composed of particles of dolomite, basalt, dacites, andesites, rhyolites, sandstone, quartz, and quartzite. The properties of the asphalt binder and aggregates used are shown in Table 2.

A dense asphalt mixture type IV-12 according to Chilean specifications was designed as the reference mixture (HMA/R) in the study. The design of the HMA/R and the determination of the optimum asphalt content (OAC) were carried out using the Marshall Design Method, fulfilling Chilean specifications. The OAC of 5.3% was obtained for the HMA/R reference mixture and met the requirements for stability (kN), flow (0.25 mm), air voids (percent), and voids in mineral aggregate (percent).

#### 2.2.2. Selection of Three Percentages of Additive to Evaluate

The selection of the addition percentages of the developed additive to be evaluated in the asphalt mixture was made based on the analysis carried out in the literature review shown in Table 1. In this analysis, the optimal addition content of different types of fibers was determined according to their effect on important mechanical properties of asphalt mixtures. The results obtained from those studies that evaluated the main components of the WTTF, which were mostly thermoplastic polymers such as polyester (polyethylene terephthalate: PET) and polyamides (nylon), were prioritized. Therefore, for this experimental study, the additions of 2%, 5%, and 8% (by weight of asphalt binder) of the WTTF additive were defined, as shown in Figure 5. Moreover, the design properties were verified for each percentage of WTTF additive selected, fulfilling Chilean specifications, as shown in Table 3. The OAC of 5.3% was used in the manufacture of all evaluated asphalt mixtures. The OAC considers a tolerance value of ±0.01%. The cationic asphalt emulsion contained in the WTTF-based additive, and added in the asphalt mixture, does not affect the OAC because its content remained under the accepted tolerance. For the asphalt mixtures’ manufacture, first, the aggregates and the liquid asphalt binder were heated at 154 °C (according to the used AB specification) and weighed separately, then, the WTTF-based additive was weighed according to the addition percentage to evaluate, and finally, the aggregates, the asphalt binder, and the WTTF-based additive were mixed.

#### 2.2.3. Testing Methods

Mechanical or performance properties evaluated in the mixtures made in the third stage of the experimental plan were stiffness modulus, moisture sensitivity, permanent deformation, and cracking resistance.

The stiffness modulus (S_M_) of the specimens was determined using the indirect tensile strength test in accordance with European standard UNE-EN 12697-26 (annex C). This methodology consists of applying a sinusoidal pulse of load and rest period to produce a controlled horizontal deformation in cylindrical specimens. Each test consisted of 15 load cycles, applying a vertical compression load for 0.124 ms. To calculate the stiffness modulus, only cycles from 10 to 15 were used, as the previous one is considered as conditioning of the specimen. Five cylindrical specimens were manufactured per type of asphalt mixture evaluated, with a total of 20 specimens tested, and an average result for each type of asphalt mixture was analyzed. The specimens were kept at 20 °C for 24 h before testing. The stiffness modulus (*S_M_*) is obtained from Equation (1).
(1)$$S_{M}=\frac{F\cdot \left(v+0.27\right)}{\left(z\cdot h\right)}$$
where *S_M_* is the stiffness modulus measured in MPa, *F* is the maximum vertical load applied in N, *v* is Poisson’s ratio, z is the horizontal displacement in mm, and *h* is the average thickness of the specimen in mm.

Moisture sensitivity (or moisture damage) was analyzed by means of the indirect tensile strength ratio (ITSR) in accordance with European standard UNE-EN 12697-12. This method uses the indirect tensile strength (ITS) parameter of dry and wet specimens in accordance with European standard UNE-EN 12697-23. The procedure consists of the manufacturing of six cylindrical specimens per type of asphalt mixture evaluated (using a Superpave gyratory compactor) and dividing them into two equivalent subgroups of three specimens with similar physical characteristics. One of these subgroups is used to determine the mean ITS parameter under dry conditions, whereas the other subgroup is used to determine the ITS parameter under wet conditions. The dry condition consists of air conditioning the specimens at 20 ± 5 °C, whereas the wet condition consists of subjecting the specimens to saturation in water by applying a vacuum at a pressure of 6.7 kPa for 30 ± 5 min and then keeping them immersed in a water bath at 40 °C for a period between 68 and 72 h. Then, both subgroups of specimens are tested under indirect tensile strength at 25 °C. Later, the indirect tensile strength ratio (ITSR) is obtained from Equations (2) and (3).
(2)$$ITS=\frac{2\cdot P}{\pi \cdot D\cdot H}\cdot 100,$$
(3)$$ITSR=\frac{ITS_{w}}{ITS_{d}}\cdot 100,$$
where *ITS* is the indirect tensile strength (kPa), *P* is the maximum applied load (kN), *D* is the specimen diameter (mm), *H* in the specimen height (mm), *ITS_w_* is the mean resistance to indirect tensile strength of the wet-conditioned test samples (Pa), and *ITS_d_* is the mean resistance to indirect tensile strength of the dry-conditioned test samples (Pa).

Rutting resistance was evaluated by means of the Hamburg Wheel Tracking Test (HWTT) in accordance with standard AASHTO T 324. This method determines the susceptibility to permanent deformation and stripping effects of asphalt mixture specimens immersed in a water bath at 50 ± 0.5 °C. This test provides information about the rate of permanent deformation from a constant load of 705 N applied to the specimens for 10,000 cycles (or 20,000 passes). Performance parameters such as mean rut depth (RD), slope of the strain curve between cycles 5000 and 10,000 (WTS), and percentage of mean rut depth (PRD) can be determined. In addition, this test evaluates signs of moisture damage through the loss of adhesiveness between asphalt binder and aggregates (stripping) due to the combined effects of water and load. If this phenomenon occurs, an increase in the slope of the deformation curve can be observed. Four specimens per type of asphalt mixture were manufactured and tested.

Cracking resistance of the evaluated mixtures was analyzed by the Fenix test. This test simulates mode I of fracture, which is the main cracking mechanism of an asphalt mixture [35]. This test is used to measure the dissipated energy during the cracking process together with tensile stress and displacement parameters. The Fenix test generates tensile stresses around the induced crack, using the work performed to propagate the crack across the fracture plane. The test procedure consists of subjecting one half of a cylindrical specimen to a tensile stress at a constant displacement velocity of 1 mm/min and specific temperature. A 6 mm-deep notch is made in the middle of its flat side where two steel plates are fixed. Each plate is attached to a loading platen so that they can rotate about fixing points. Load and displacement data are recorded throughout the test to calculate the mechanical parameters involved in the cracking. Parameters such as maximum tensile force (F_max_), maximum strength (R_T_), displacement at 50% of post-maximum tensile force (d_50PM_), and displacement between d_50PM_ and d_M_ (D_T_) can be determined from the load–displacement output curve, whereas the dissipated energy per unit area (G_D_) parameter and the toughness index (*T_I_*) can be determined by Equations (4) and (5), respectively. G_D_ is related to the work performed divided by the fracture area, and the *T_I_* is related to the capacity of the asphalt mixture to hold its components together once their maximum resistance is reached.
(4)$$G_{D}=\frac{\int_{0}^{\mathrm{df}}F\left(x\right)\cdot \mathrm{dx}}{S},$$
where G_D_ is the dissipated energy during cracking process in J/m^2^, F is the load in kN, x is the displacement in mm, d_f_ is the fracture displacement in mm, and S is the fracture area in m^2^.
(5)$$T_{I}=\frac{\int_{\mathrm{dM}}^{\mathrm{df}}\;F\left(x\right)\cdot {\mathrm{dx}}_{\;}}{S}\times \left(d_{50\mathrm{MP}}-d_{M}\right),\;$$
where *T_I_* is the toughness index in J mm/m^2^, d_M_ is the displacement at F_max_ in mm, d_50MP_ is the displacement after maximum load at ½ F_max_ in mm, and S is the fracture area in m^2^.

Three specimens for each evaluated mixture were manufactured and tested, considering two temperatures: 0 °C and 10 °C.

#### 2.2.4. Statistical Analysis

A statistical analysis was performed to verify the significance of the results obtained from the experimental tests, avoiding analytical errors due to dispersion of the data. The null hypotheses of normality and homoscedasticity were tested separately for each set of data in the different experimental tests, using the Shapiro–Wilk test and Levene’s test, respectively. This makes it possible to choose the type of statistical test to be used in each case. A significance level of 95% was considered for all tests carried out. The parametric test used for data with normal distribution was the analysis of variance (ANOVA), evaluating the *p*-value and the Fisher–Snedecor F statistic. Friedman, Kendall’s W, and Wilcoxon signed rank tests were the non-parametric tests used for data that did not follow a normal distribution.

## 3. Results and Discussion

The average results for the stiffness modulus at 20 °C and for the densities of the evaluated mixtures are shown in Figure 6. According to these results, the density values recorded in the mixtures with different percentages of additive were similar to those of the HMA reference mixture (HMA/R). Regarding the stiffness modulus, the mixtures with 2% (HMA/2) and 5% (HMA/5) of additive content were higher than the mean value of the HMA/R by 13% and 10%, respectively. The latter becomes important for asphalt mixtures since it improves the structural capacity of the pavement [36]. The addition of 2% and 5% of additive content allows a good distribution of the WTTF in the mixture, favoring its dispersion in a three-dimensional network. According to studies that applied other fibers, such as steel fibers, glass fibers, and other synthetic fibers, this three-dimensional network improves the stiffness of the asphalt mixture, increasing internal cohesion forces [14,15,37]. Another aspect that can influence the observed effect is the good adhesion that occurs between the fiber and the asphalt binder, improving the mechanical behavior of the asphalt mastic at the temperature evaluated, as indicated by Calabi-Floody et al. (2019) and Chen and Lin (2005) [38,39]. On the other hand, the mixture with the addition of 8% of additive content (HMA/8) recorded an average stiffness modulus value similar to the HMA reference mixture. The cause of this effect may be because the excess of WTTF could cause a poor distribution of the fibers in the mixture. This result agrees with what was reported by Bocci and Prosperi (2020), who indicated that for 6% of WTTF addition there is no effect on the stiffness modulus property at 20 °C, obtaining values similar to those of the reference mixture [6].

The statistical analysis of the data obtained from the stiffness modulus showed a non-parametric behavior. The Friedman and Kendall’s W tests applied to the data showed that there was a significant difference in the mean of the stiffness modulus for all the mixtures evaluated (*p*-value = 0.024 < 0.05). Additionally, the results obtained by the Wilcoxon signed rank test indicated that the mixtures with the addition of 2% and 5% of additive content had a significant difference (*p*-value = 0.028) in the values obtained for the stiffness modulus compared to the HMA/R. By contrast, the same test indicated that the addition of 8% of additive content did not differ significantly (*p*-value = 0.173) from the HMA/R.

Figure 7a shows the results obtained from the average values for the indirect tensile strength of the mixtures in dry (ITS_d_) and wet (ITS_w_) conditions together with the values of the densities of the evaluated mixtures. The density values recorded in the mixtures with different additive content were lower than those of HMA/R for the same level of compaction (40 gyrations). This indicates that the higher the content of the WTTF-based additive, the higher the compaction energy required in the mixtures. These results are consistent with what was described by the study carried out by Bocci and Prosperi (2020) [6]: that the presence of WTTF fibers reduces the compactability of the mixture, which directly affects the content of air voids [40]. In relation to the indirect tensile strength, the mixtures with the addition of 2% and 5% of WTTF-based additive did not present statistically significant differences (*p*-value > 0.05) in the individual ITS tests (dry and wet) compared to the HMA reference mixture. However, the mixture with the addition of 8% of WTTF-based additive registered a significant reduction (*p*-value < 0.05) in the ITS value in the ITS_d_ and ITS_w_ conditions, with average values of 18% and 24%, respectively. This effect could be caused by the reduction in the HMA/8 density evaluated in relation to the HMA/R, which affects the indirect tensile strength of the mixture, as pointed out by Moreno-Navarro et al. (2014) [41].

Additionally, Figure 7b shows that the mixtures with the addition of 2%, 5%, and 8% of WTTF-based additive registered a good performance of the indirect tensile strength ratio (ITSR), with values higher than those of the reference mixture and according to the limits required by European standards for base and intermediate layers (ITSR > 80%) and for wearing courses (ITSR > 85%) [42]. These results indicated that the WTTF-based additive has a positive effect on moisture susceptibility for contents of 2% and 5%, despite requiring higher compaction energy to achieve a similar reference mixture density. In other words, the use of the developed additive does not affect the adhesion and internal cohesion capacity in the aggregate-binder matrix of the mixture, reducing the possibility of causing stripping. These results agree with different studies that addressed the addition of fibers of different polymeric origin in asphalt mixtures, where it was observed that these fibers have a positive effect on the moisture damage property [15,18].

The evolution of the rut depth (RD) with respect to the number of wheel passes obtained from the Hamburg Wheel Tracking Test is shown in Figure 8. The results indicated that the mixtures with WTTF-based additive registered lower RD values than the reference mixture. The mixtures with an addition of 2%, 5%, and 8% of WTTF-based additive showed a decrease of 34%, 37%, and 29%, respectively, in the RD parameter. These results were consistent with the studies carried out by Kim et al. (2018), Taherkhani et al. (2017) and Tapkin (2008) [17,18,19], which indicated that mixtures modified with polypropylene, polyester, and nylon fibers showed lower values in the parameters related to the permanent deformation compared to the reference mixture used in their respective studies. This effect is directly related to the formation of the three-dimensional network generated by the fibers in the fiber-matrix bonding properties of the asphalt mixture, indicating that this characteristic has an important effect on the resistance to permanent deformations, as evaluated by the RD parameter [14,15,37]. The mixture with the addition of 5% WTTF-based additive showed the highest reduction rate, with a maximum difference of 8.5 mm in depth compared to the reference mixture, whereas the mix with 2% WTTF-based additive showed the lowest strain slope (Wheel Tracking Slope = WTS) between 5000 and 10,000 cycles. Additionally, no mixture evaluated showed a loss of adhesiveness between the stone aggregates and the asphalt binder (stripping), exhibiting good behavior against moisture damage, which agrees with the results obtained from the ITSR parameter.

Regarding the percentage of air voids established by the regulations (7 ± 1%), it is important to note that to reach the required percentage, the mixtures with 2%, 5%, and 8% of WTTF-based additive required 1.5, 2.5, and 3.5 times more compaction energy, respectively, compared to the reference mixture, to achieve the same level of densification. This phenomenon can be attributed to the fact that the use of fibers in asphalt mixtures reduces the compatibility of the mixture [6]. This agrees with what is illustrated in Figure 7a, where it is observed that as the additive content increases, the density value for the same compaction energy is reduced.

The statistical analysis showed a normal distribution and variance homogeneity in the data obtained from the HWTT, according to the Shapiro–Wilk (*p*-value = 0.741) and Levene’s (*p*-value = 0.640) tests. The ANOVA indicated that the content of WTTF significantly influences the RD parameter with a 95% confidence (*p*-value <0.05), and a sufficiently high F-ratio to accept that the mixtures are statistically different in the performance of this property. The coefficient of determination (R2 ≈ 98%) showed a good fit to the study variables to explain the RD parameter.

The parameters of the Fenix test related to cracking resistance of the evaluated asphalt mixtures are shown in Figure 9. The effect of the additive on the maximum tensile force (Fmax) and on the bending capacity of the mixture (d50PM) is observed in Figure 9a. At a temperature of 0 °C, the mixtures with 2% and 5% of WTTF-based additive showed a Fmax similar to the reference mixture, but a decrease in the d50PM parameter. However, for mixes with 8% of WTTF-based additive, a lower Fmax and a higher bending capacity than the reference mixture was observed. Moreover, at 10 °C, all the evaluated mixtures with WTTF-based additive showed a similar behavior in the ability to resist tensile stress and higher values in the bending capacity than the reference mixture. The results obtained for the energy dissipated in the cracking process (GD) and the toughness index (TI) can be seen in Figure 9b. Regarding the dissipated energy, similar values were observed between the mixtures with different contents of WTTF-based additive for both temperatures evaluated. However, at 0 °C, the mixtures with different contents of WTTF-based additive showed an average decrease of 11% in relation to the reference mixture, whereas at 10 °C they presented an average increase of 17% in the GD parameter. The effect of the WTTF additive on the toughness index (TI) at a temperature of 0 °C showed a similar behavior to the reference mixture for the contents of 5% and 8% of WTTF-based additive, with a slight decrease for the 2% content. Conversely, at 10 °C, all the percentages of additive content evaluated showed an average increase of 79% in the TI value in respect to the reference mixture. These results showed that the WTTF-based additive has a greater effect at a temperature of 10 °C, indicating that the mixtures require more work to produce cracking, and, at the same time, the mixtures showed greater toughness for having reached their maximum strength.

Figure 9c shows the Fenix Stress-Strain Diagram©, which is an established procedure for the characterization of bituminous mixtures according to two important parameters of the mixture to define its behavior [43]. According to the results, the response of the mixtures with the WTTF-based additive showed a greater increase in its ductility between the evaluated temperatures of 0 °C and 10 °C. At 10 °C, an improved ductility was observed compared to the reference mixture, for similar maximum strength values.

The statistical analysis showed a normal distribution and variance homogeneity in the data obtained from the Fenix assay, according to the Shapiro–Wilk and Levene’s tests (both *p*-values > 0.05). In this context, an ANOVA was performed that evaluated the influence of the temperature and WTTF-based additive content as variables on the main parameters of the Fenix assay. The results indicated that the evaluated variables showed significant differences (*p*-value < 0.05) for each of the parameters studied. The calculated coefficients of determination were around 80% and 90%.

## 4. Conclusions

The WTTF-based additive can be used for asphalt mixtures, improving their properties related to performance in the pavement, complying with design specifications. Based on the results obtained in the properties related to the performance and the statistical analysis, the following conclusions can be drawn:The physical chemical analyses show that the main fiber that composes the WTTF evaluated was polyester. According to the melting temperatures measured and the literature reviewed, polyester is a suitable polymer to be used in asphalt mixtures that require a processing temperature range between 140 °C and 190 °C, since its melting and decomposition point is higher.In relation to the stiffness property at 20 °C, the use of the WTTF-based additive up to 5% registered a significant increase for the stiffness modulus value compared to the reference mixture. However, for higher addition percentages, no significant difference was observed for the stiffness modulus value in relation to the reference mixture.The use of a WTTF-based additive in asphalt mixtures requires a higher compaction energy to achieve the same level of densification as the reference mixture, which is evidenced by the samples compacted through the Superpave gyratory compactor for the equal compaction energy (moisture sensitivity tests) and for the equal air void percentage required (rutting resistance tests).Regarding the results of the moisture sensitivity test, none of the mixtures evaluated that used the WTTF-based additive showed a significant effect in the ITS parameter (dry and wet). However, all the mixtures meet the ITSR limits required by European standards for the different asphalt layers of a pavement structure.The resistance to permanent deformation shows that the use of the WTTF-based additive has a significant effect on the rut depth (RD) parameter, with reductions between 30% and 37% compared to the reference mixture, and with lower WTS. Moreover, none of the mixtures evaluated showed stripping, exhibiting good behavior against moisture damage.The cracking resistances of the mixtures that used the WTTF-based additive at 10 °C show that more work is required to produce cracking (GD) and they register a higher toughness (TI). However, the mixtures show a higher stiffness at 0 °C, with higher values in maximum resistance (Fmax) but lower levels of bending capacity (d50PM).The WTTF is currently considered a massive by-product from the ELT recycling industry; however, in this study it was successfully used in the development of a new additive to improve the performance properties of the asphalt mixture. The greatest effects were on the properties of rutting resistance, stiffness modulus, and moisture damage.

## Figures and Tables

**Figure 1 polymers-14-03250-f001:**
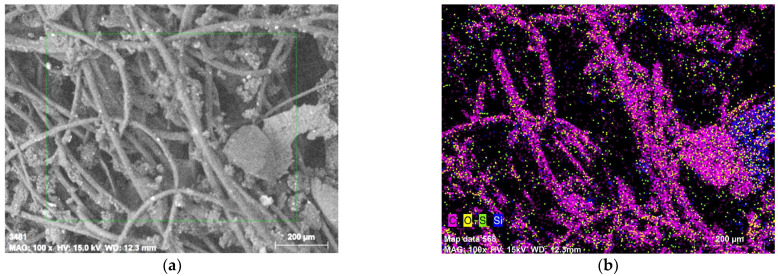
SEM-EDX images from WTTF used in this study. (**a**) Original SEM image, and (**b**) elemental distribution detected in the image.

**Figure 2 polymers-14-03250-f002:**
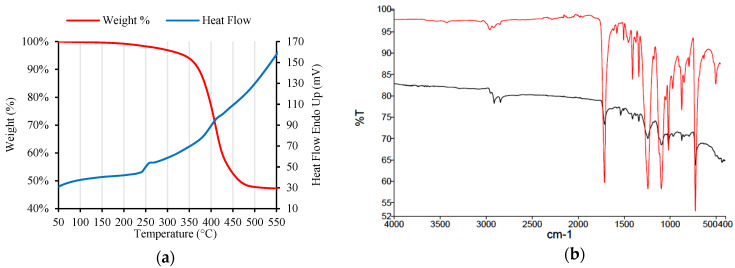
(**a**) Temperature profile from TGA of WTTF; (**b**) FTIR spectrum from WTTF sample (red) and a polyester standard (black).

**Figure 3 polymers-14-03250-f003:**
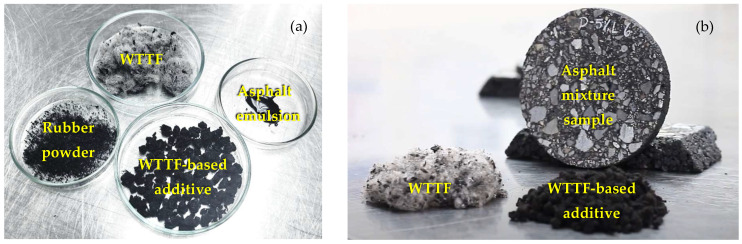
Images of the WTTF-based additive with its components (**a**) and its application on a sample (**b**).

**Figure 4 polymers-14-03250-f004:**
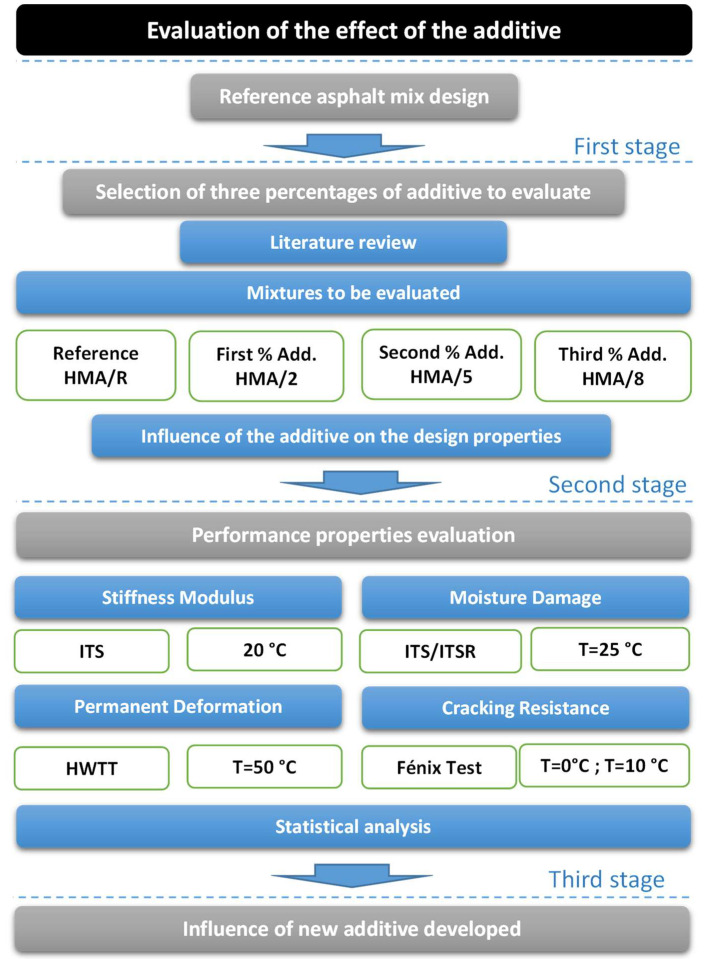
Experimental scheme of the study to show the effect of the new additive based on WTTF.

**Figure 5 polymers-14-03250-f005:**
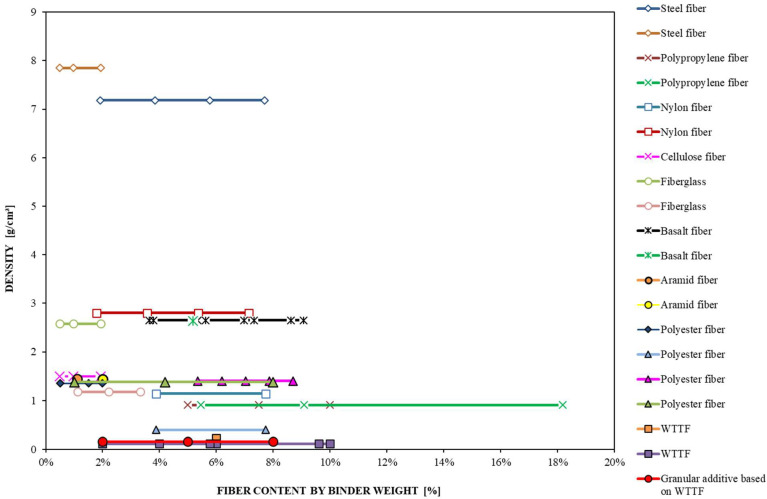
Fiber contents by weight of asphalt binder (literature review).

**Figure 6 polymers-14-03250-f006:**
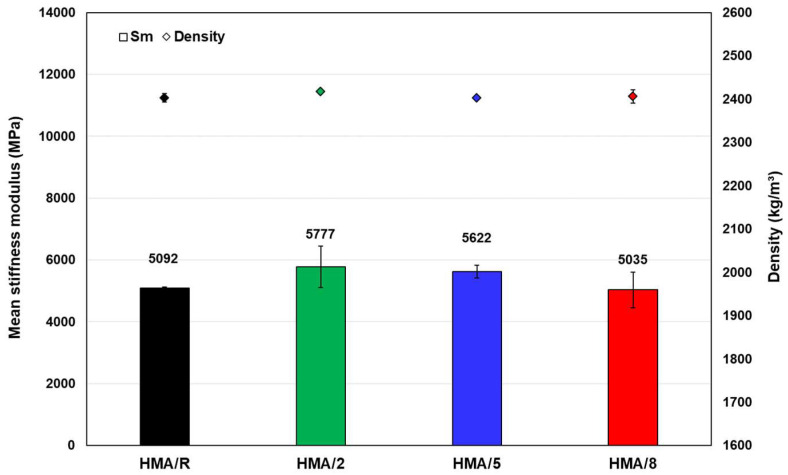
Stiffness modulus at 20 °C for all the studied mixtures.

**Figure 7 polymers-14-03250-f007:**
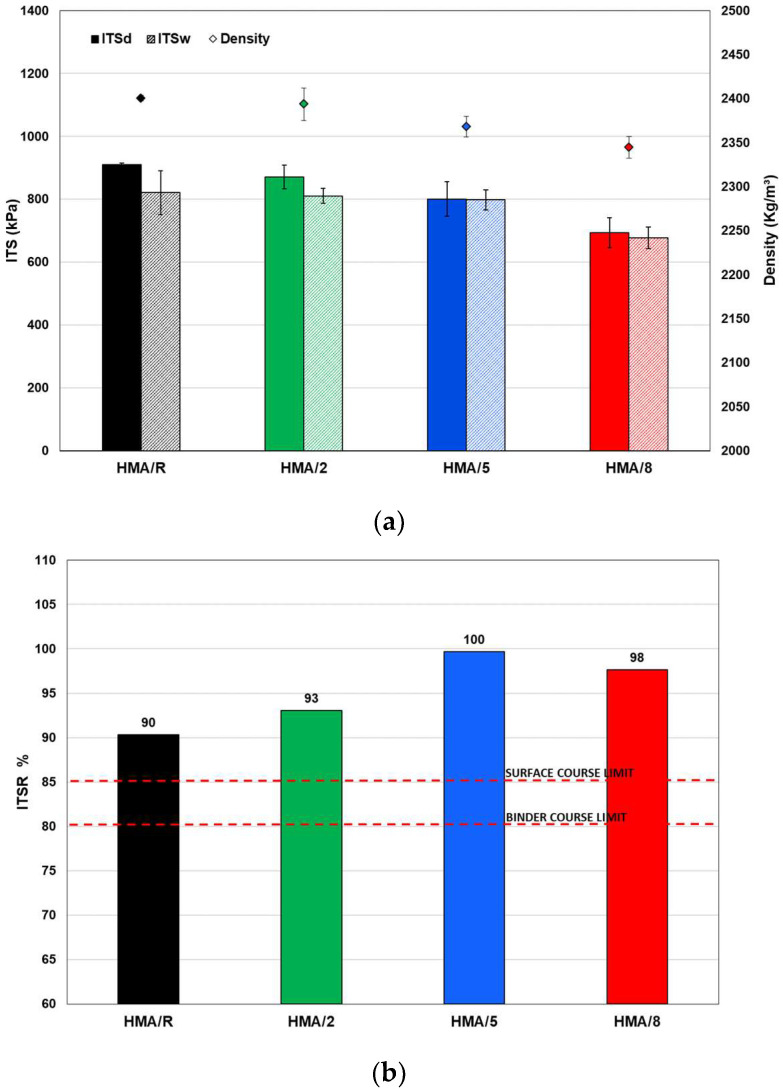
(**a**) Indirect tensile strength of dry and wet samples; (**b**) Indirect tensile strength ratio, for all the studied mixtures.

**Figure 8 polymers-14-03250-f008:**
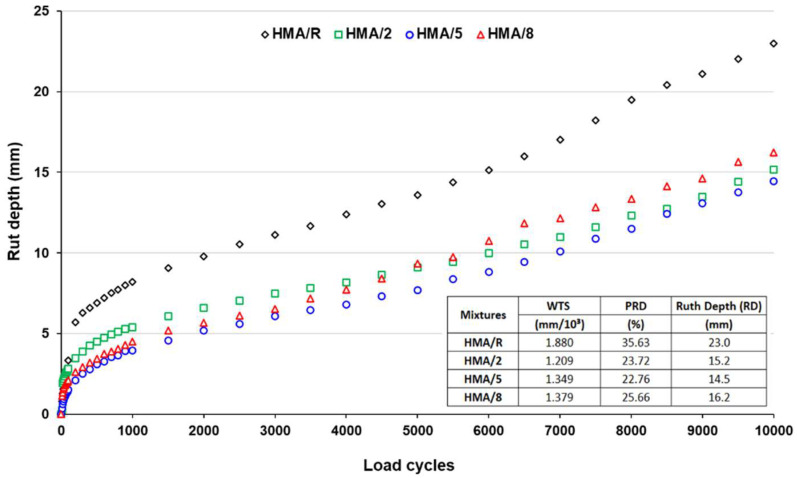
Hamburg Wheel Tracking Test results at 50 °C.

**Figure 9 polymers-14-03250-f009:**
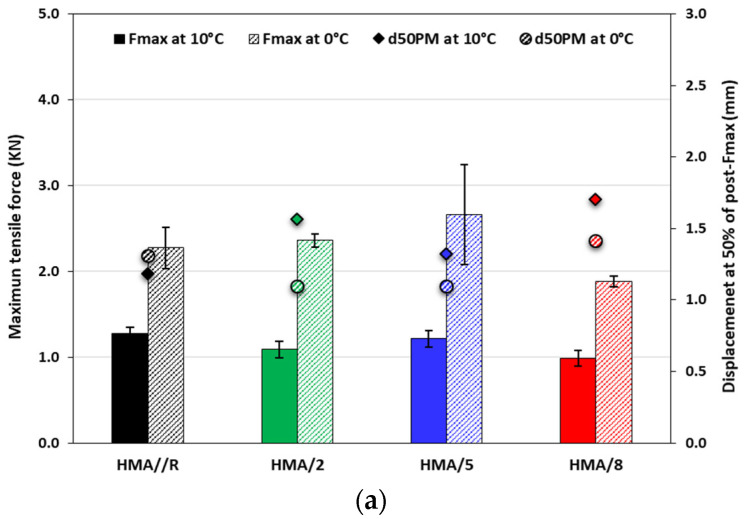
Fenix test parameters: (**a**) Maximum tensile force vs. Displacement at 50% of post-Fmax; (**b**) Dissipated energy vs. Toughness Index; (**c**) Fenix Stress-Strain Diagram©. Temperatures 0 °C and 10 °C. (The Fenix diagram is associated with a test methodology. 2019 UPC all Rights Reserved)

**Table 1 polymers-14-03250-t001:** Different fibers applied in asphalt mixes.

Fiber Type	Ref.	Density (ρ); Length (L); Diameter (Ø)	Added Amount		Advantages	Disadvantages
			(By Binder Weight)	(By Binder Volume)		
**Steel**	[14]	ρ: 7.18 g/cm^3^ L: 2–8 mm Ø: 0.157 mm	1.9% 3.8% 5.8% 7.7%	0.3% 0.6% 0.8% 1.1%	Good distribution in the asphalt mix.	Susceptible to the impact of the aggregates during the mixing and compaction process, since they are thick fibers and of medium length.
	[15]	ρ: 7.85 g/cm^3^ L: 4–12 mm Ø: 0.18–0.3 mm	0.5% 1.0% 1.9%	0.06% 0.1% 0.3%	Good distribution in the mixture, forming a three-dimensional network. Increases in resistance to indirect traction.	Less resistance to fatigue with a content of 2%.
Polypropylene	[16]	ρ: 0.91 g/cm^3^ L: 12 mm	5.0% 7.5% * 10.0%	5.7% 8.5% 11.3%	Increases in the apparent density and in the Marshall stability of asphalt mixtures.	
	[17]	ρ: 0.91 g/cm^3^ L: 3–50 mm	5.5% 9.1% 18.2% *	6.2% 10.3% 20.7%	Increases in the Marshall stability of the mixes. Improvement in the properties of permanent deformation and resistance to fatigue.	The fibers have a different specific density and size distribution, which can cause differences in the behavior of the mixtures.
	[18]	ρ: 0.91 g/cm^3^ L: 6 mm Ø: 40 μm	3.9% * 7.7%	4.4% 8.8%	Improves Marshall stability, indirect tensile strength (ITS), and moisture susceptibility.	Most of the fibers dissolve during the modification process due to their low melting point. Over 9% of the fiber, the mechanical properties of the mixture are affected, except for permanent deformation and toughness.
Nylon	[18]	ρ: 1.14 g/cm^3^ L: 12 mm Ø: 23 μm	3.9% 7.7% *	3.5% 7.0%	Improvement in the mechanical properties of the asphalt mix: Marshall stability, indirect tensile strength (ITS), permanent deformation, and bending capacity.	
	[19]	ρ: 1.4 g/cm^3^ L: 12 mm Ø: 20 μm	1.8% 3.6% 5.4% 7.1% *	0.6% 1.3% 1.9% 2.6%	Increases Marshall stability. Improves resistance to permanent deformation. Improves resistance to fatigue cracking.	
Cellulose	[15]	ρ: 1.50 g/cm^3^ L: 0.02–2.5 mm Ø: 0.025 mm	0.5% 1.0% 1.9%	0.3% 0.7% 1.3%	Increases in resistance to indirect traction. High resistance to moisture damage. Improved resistance to low temperature cracking	
Fiberglass	[15]	ρ: 2.58 g/cm^3^ L: 6–13 mm Ø: 0.012–0.02 mm	0.5% 1.0% 1.9%	0.2% 0.4% 0.8%	Increases in indirect tensile strength. High resistance to moisture damage. Improves resistance to low temperature cracking.	Less resistance to fatigue with a content of 2%.
	[20]	ρ: 1.18 g/cm^3^ L: 12 mm Ø: 0.13 mm	1.1% 2.2% 3.3%	1.0% 1.9% 2.9%	Improves resistance to cracking with a content of 1% and 2%. The positive effect of the fibers outweighs the adverse impact of the reclaimed asphalt pavement material.	About 2% the resistance to cracking of the mixtures is affected. However, a better performance is maintained compared to the control mixture.
Aramid	[21]	ρ: 1.44 g/cm^3^ L: 6 mm	1.1%	0.8%	Fiber positively influences abrasion resistance.	Decrease in the content of air voids. Lower resistance to indirect tensile strength (ITS). Increased moisture susceptibility.
	[22]	ρ: 1.44 g/cm^3^ L: 1–6 mm Ø: 12 μm	2%	1.4%	Improves performance at high temperatures. Improves the viscosity of the modified asphalt cement, resulting in an increase in the modulus of rigidity.	Agglomeration of the fibers with a length of 6 mm. Fibers do not provide a promising effect on low temperature crack resistance.
Basalt	[23]	ρ: 2.64 g/cm^3^ L: 9 mm Ø: 13–15 μm	3.6% 5.3% * 7.0% * 8.6% 3.8% 5.6% 7.3% 9.1%	1.4% 2.0% 2.7% 3.3% 1.5% 2.2% 2.8% 3.5%	Good behavior at low temperatures. The fibers form a network-like structure, which allows for improved integrity, disperses stress, and delays the spread of microcracks.	Fiber dispersion is not uniform at high contents.
	[24]	ρ: 2.63 g/cm^3^ L: 24 mm Ø: 18 μm	5.2%	2.0%	The fibers are highly effective in improving the properties of blends at low temperatures. Fibers have a positive effect on fatigue behavior.	The fibers alone are not effective in improving the useful life of the mix compared to the control mix.
Polyester	[25]	ρ: 1.35 g/cm^3^ L: 10–20 mm Ø: 30 μm	0.5% 1.0% * 1.5% 2.0%	0.4% 0.7% 1.1% 1.5%	Improves resistance to fatigue.	Due to the agglomeration that is generated in the mixture, the maximum amount of fiber content is restricted to 2%.
	[18]	ρ: 1.40 g/cm^3^ L: 6 mm Ø: 41 μm	3.9% 7.7% *	2.9% 5.7%	Improvement in the mechanical properties of the asphalt mix: Marshall stability, indirect tensile strength (ITS), permanent deformation, and bending capacity.	
	[26]	ρ: 1.4 g/cm^3^ L: 12 mm Ø: 20 μm	5.3% 6.2% 7.0% * 7.9% 8.7%	4.0% 4.6% 5.3% 5.9% 6.5%	Improves resistance to low temperature cracking.	The distribution of the fiber in the mix is affected about 7%.
	[27]	ρ: 1.38 g/cm^3^ L: 4–24 mm Ø: 20 μm	1.0% 4.2% * 8.0%	0.8% 3.2% 6.0%	Improves tensile deformation properties. Maintains tensile ductility, independent of temperature change (low temperatures).	A fiber content of less than 1% and more than 8% has an unfavorable effect on tensile performance. The fiber length is crucial to obtain a good performance and distribution in the mix. Optimal length 6 mm.
WTTF	[6]	ρ: 0.18 g/cm^3^ L: 1–2.5 mm Ø: 5–40 μm	6.0%	34.5%	Improved resistance to fatigue. Maintains modulus of rigidity at low / intermediate temperatures, relative to the control mix. After failure, the fibers create a sealing effect at the edges of the cracks and contrast the fracture.	The presence of WTTF reduces the compactness of the mix. Therefore, it is recommended to increase the filler content by about 2% by weight.
	[8]	ρ: 0.17 g/cm^3^ Ø: 22.5 μm	2.0% 4.0% 6.0% * 5.8% 10.0% 9.6%	11.8% 23.7% 35.5% 34.1% 59.2% 56.9%	Increases in void content. Increases in indirect tensile strength (ITS). Its use is a promising scenario to reduce environmental impact. The high economic return makes this recycling system economically sustainable.	

* Optimum value.

**Table 2 polymers-14-03250-t002:** Properties of the asphalt binder and aggregates used in the tests.

Properties	Results	Specs [31]
**Asphalt binder CA-24**		
Absolute viscosity at 60 °C, 300 mm Hg (P)	3072	Min 2400
Penetration at 25 °C, 100 g, 5 s (0.1 mm)	58	Min 40
Ductility at 25 °C (cm)	>150	Min 100
Softening point R and B (°C)	51.4	To be reported
**Coarse aggregate**		
Specific gravity (kg/m^3^)	2685	-
Absorption (%)	1.54	-
Los Angeles abrasion loss (%)	20	Max. 25 (*)–35
Crushed aggregates (%)	96	Min. 90 (*)–70
**Fine aggregate**		
Specific gravity (kg/m^3^)	2650	-
Absorption (%)	1.1	<3
**Combined aggregate**		
Soluble salts (%)	0	Max. 2 (*)–3
Sand equivalent (%)	70	Min. 50 (*)–40

(*) Wearing course.

**Table 3 polymers-14-03250-t003:** Design parameters for design of evaluated mixes.

Mix Type	Manufacturing Temperature	Total Bitumen Content	WTTF Additive	Density	Stability	Flow	Air Voids	VMA
	(°C)	(% by Weight of Aggregate)	(% by Weight of AB)	(kg/m^3^)	(N)	0.25 mm	(%)	(%)
HMA/R	154	5.30	0	2418	13,745	10.8	3.1	13.9
HMA/2	154	5.30	2	2408	13,471	10.9	3.4	14.2
HMA/5	154	5.30	5	2411	14,414	11.5	3.3	14.1
HMA/8	154	5.30	8	2415	13,947	10.6	3.2	14.0
Chilean Specifications [31,34]			Surface course		>9000	8–14	3–5	>13

## Data Availability

Not applicable.

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
