# Peer review of "Development of a New Additive Based on Textile Fibers of End-of-Life Tires (ELT) for Sustainable Asphalt Mixtures with Improved Mechanical Properties"

_polymers, 2022, doi:10.3390/polym14163250_

Round 1
Reviewer 1 Report
This review report has been removed from the review record as it did not conform with MDPI’s standards (https://www.mdpi.com/reviewers#_bookmark11).
Reviewer 2 Report
The article concerns an important, from the point of view of environmental protection, problem of the management of polymer fibers from end-of-life tires. The Authors of the article, based on literature reports, proposed the use of this waste as a component of hot asphalt mix. In the introduction of the reviewed article the Authors presented the state of knowledge, that they described on the basis of current sources, mostly from 2016-2021. Information contained in the literature review The Authors used, inter alia, for determine the composition of the asphalt binder with the addition of waste tire textile fibers. Binders containing three different amounts of waste tire textile fibers (2%, 5% and 8%) were used to obtain asphalt mixtures. Asphalt mixtures related to their performance in pavement were tested. Their stiffness modulus, moisture sensitivity, rutting resistance and cracking resistance were determined. Statistical analysis of the results was performed. The presentation of the results and their discussion are very clear and transparent.
In my opinion, the article may be published in POLYMERS in the form presented by the Authors.
Author Response
Response to Reviewer 2 Comments
Point 1: The article concerns an important, from the point of view of environmental protection, problem of the management of polymer fibers from end-of-life tires. The Authors of the article, based on literature reports, proposed the use of this waste as a component of hot asphalt mix. In the introduction of the reviewed article the Authors presented the state of knowledge, that they described on the basis of current sources, mostly from 2016-2021. Information contained in the literature review The Authors used, inter alia, for determine the composition of the asphalt binder with the addition of waste tire textile fibers. Binders containing three different amounts of waste tire textile fibers (2%, 5% and 8%) were used to obtain asphalt mixtures. Asphalt mixtures related to their performance in pavement were tested. Their stiffness modulus, moisture sensitivity, rutting resistance and cracking resistance were determined. Statistical analysis of the results was performed. The presentation of the results and their discussion are very clear and transparent.
In my opinion, the article may be published in POLYMERS in the form presented by the Authors.
Response 1: The authors thanks to Reviewer 2 for the positive evaluation.

Reviewer 3 Report
This study applied an additive made with waste tire textile fibers (WTTF), which was incorporated at various proportions in conventional hot mix asphalt (HMA) for pavement properties investigations. The topic is interesting, but there are still many existing questions needing to be carefully addressed. Upon this, I suggest having this draft paper carefully revised prior to publication.
- Section 2: it is no need to discuss much of raw materials in Materials and Methods; check if Figure 2 is necessary to present.
- Figure 3, there is no specific label for readers to clearly understand the items. It is also suggested that the numberings like (a), (b), (c), etc. can be used for pictures/figures combined in one figure.
- Section 2.3: a more detailed description of the mixing of WTTF additive to HMA should be provided. It also needs to update the research flowchart (Figure 4) to a high resolution one.
- Line 200-201: It should state whether the cationic asphalt emulsion will affect with OAC during mix mixing.
- It would be better to correct sections “3. Results” to “3. Results and Discussion”, as well as “4. Discussion” and “5. Conclusions” to “4. Conclusions”.
- Figure 8: it should justify why the rut depth of HMA first decreases and then increases as the incorporation of WTTF additive increases from 2% to 8%.
- Figure 9 (b): explain why the GD value of HMA/8 at 0 ℃ is higher than that of HMA/5 and HMA /2.
- Conclusion Section: the influence of WTTF additive content on the performance properties of the mixes should be supplemented.
- It needs double check the expressions and logics in the context.
- For Introduction Section, a brief background introduction of the application of ELT recycled into asphalt pavement needs to be enhanced. Some literatures, like “sustainable practice in pavement engineering through value-added collective recycling of waste plastic and waste tyre rubber” can be taken into account.
Author Response
Response to Reviewer 3 Comments
Point 1: Section 2: it is no need to discuss much of raw materials in Materials and Methods; check if Figure 2 is necessary to present.
Response 1: As the Reviewer 3 suggested, the authors revised the content of the section Materials and Methods and considered that the Figure 2 is important due to it shows the melting point in the TGA and confirms the polyester composition in the FTIR analysis of the WTTF. The WTTF melting point identified makes this fiber suitable for hot mix asphalt manufacturing, between 140° and 190° C.
Point 2: Figure 3, there is no specific label for readers to clearly understand the items. It is also suggested that the numberings like (a), (b), (c), etc. can be used for pictures/figures combined in one figure.
Response 2: As the Reviewer 3 suggested, the authors added labels in the Figure 3 which identify the components and the application of the WTTF, and the numbering (a) and (b).
Point 3: Section 2.3: a more detailed description of the mixing of WTTF additive to HMA should be provided. It also needs to update the research flowchart (Figure 4) to a high resolution one.
Response 3: As the Reviewer 3 suggested, the authors added the information of the mixing procedure in the lines 215 to 219. Also, the Figure 4 was changed by one with high resolution.
Point 4: Line 200-201: It should state whether the cationic asphalt emulsion will affect with OAC during mix mixing.
Response 4: As the Reviewer 3 suggested, the authors added a statement which indicates that the cationic asphalt emulsion does not affect hte OAC, because its content remains under the accepted tolerance (Lines 212 - 215).
Point 5: It would be better to correct sections “3. Results” to “3. Results and Discussion”, as well as “4. Discussion” and “5. Conclusions” to “4. Conclusions”.
Response 5: As the Reviewer 3 suggested, the authors change the section titles as they proposed.
Point 6: Figure 8: it should justify why the rut depth of HMA first decreases and then increases as the incorporation of WTTF additive increases from 2% to 8%.
Response 6: As the Reviewer 3 suggested, the authors described Figure 8 and stated in the manuscript that the rut depth in the asphalt mixtures with each addition percentage evaluated (2 %, 5 %, and 8 %) decreases compared with the reference. However, the slope of the rut depth curves increases upon the 6,000 cycles. This slope behavior was present either in the reference mixture as in the asphalt mixtures with the different WTTF-based additive incorporated, therefore, it is not considered that the rut depth increases with the incremental incorporation of the additive.
Point 7: Figure 9 (b): explain why the GD value of HMA/8 at 0 ℃ is higher than that of HMA/5 and HMA /2.
Response 7: As the Reviewer 3 suggested, the authors revised the Figure 9(b). Due to the dispersion of the GD results, there is no significant difference in the GD values for HMA/2, HMA/5 and HMA/8, despite the average GD value for HMA/8 is higher than HMA/5 and lower than HMA/2. Nevertheless, the GD values for HMA/2, HMA/5 and HMA/8 were lower than the HMA/R.
Point 8: Conclusion Section: the influence of WTTF additive content on the performance properties of the mixes should be supplemented.
Response 8: As the Reviewer 3 suggested, the authors revised the manuscript, and considered that the conclusions indicate the influence of the WTTF-based additive in the asphalt mixture in the different assays evaluated.
Point 9: It needs double check the expressions and logics in the context.
Response 9: As the Reviewer 3 suggested, the authors revised the redaction in the context.
Point 10: For Introduction Section, a brief background introduction of the application of ELT recycled into asphalt pavement needs to be enhanced. Some literatures, like “sustainable practice in pavement engineering through value-added collective recycling of waste plastic and waste tyre rubber” can be taken into account.
Response 10: As the Reviewer 3 suggested, the authors complemented the introduction with the literature proposed in lines 42 to 44.

Round 2
Reviewer 3 Report
The authors have already revised the manuscript carefully, and all the comments proposed are well addressed. The current version is basically satisfactory with the requirements of the Journal. Therefore, I agree to have this manuscript accepted for publication.